# Early Identification of Social, Emotional, and Behavioral Difficulties by School Staff: A Qualitative Examination of Dutch Primary School Practices

**DOI:** 10.3390/ijerph20010654

**Published:** 2022-12-30

**Authors:** Marloes L. Jaspers-van der Maten, Els W. M. Rommes, Ron H. J. Scholte

**Affiliations:** 1Department of Gender & Diversity Studies, Radboud Social Cultural Research, Radboud University Nijmegen, P.O. Box 1908, 6500 VC Nijmegen, The Netherlands; 2Behavioural Science Institute, Radboud University Nijmegen, P.O. Box 9104, 6500 VC Nijmegen, The Netherlands

**Keywords:** early identification, social emotional and behavioral difficulties (SEBD), primary schools, school staff, practices

## Abstract

About 16% of Dutch children are reported to have social, emotional, and behavioral difficulties (SEBDs). SEBDs generate distress and pose risks for various negative outcomes; thus, their timely identification is deemed important to respond appropriately to children’s needs and avoid such negative outcomes. Primary schools are considered convenient places to implement early SEBD identification, but the ways in which schools achieve this in practice may be inadequate, although the issue remains under-researched. Although there are several systematic school-based early identification methods (e.g., universal or selective screening), primary schools predominantly rely on school staff to recognize children at risk for, or experiencing, SEBDs. As differences in identification practices could impact whether and when (signs of) SEBDs are identified, this study aimed to increase our understanding of differences in identification practices used by school staff and their potential implications for early identification effectiveness. Thirty-four educational and clinical professionals working at nine primary schools participated in in-depth semi-structured interviews. We used MAXQDA to thematically code and analyze the data. Our analysis of these interviews illustrated that schools’ identification practices differed on three elements: the frequency of observations, maintaining a four-eyes principle, and the utilization of specialist knowledge. We argue that differences in these elements have potential consequences for the timeliness and quality of SEBD identification.

## 1. Introduction

Mental health difficulties are common in children and are associated with numerous negative outcomes. Approximately 8–18% of school-age children in high-income countries meet the diagnostic criteria for a mental health diagnosis, while many more will experience subclinical symptoms of distress [1]. In educational settings, the difficulties many of these children face are commonly referred to as social, emotional, and behavioral difficulties (SEBDs). Different countries, as well as adjacent disciplines in research and practice, have adopted differing terminology to describe the issues that overlapping groups of children may face (e.g., mental health difficulties, psychosocial problems, emotional and behavioral problems/difficulties/disorders). The present study focuses on practices in education; thus, in this article we have adopted the term that is commonly used in educational settings [2,3]. Although a distinction is generally made between SEBDs and clinical mental health problems, overlap between these groups of children is assumed [4]. About 16% of Dutch children are reported to have SEBDs [5], which can negatively affect educational and social engagement [6]. SEBDs also pose a risk for various negative outcomes, including worse academic performance, impaired family functioning, and longer-term mental health deterioration [3,7,8].

In order to prevent and reduce negative outcomes, countries in the Global North are making efforts to identify and respond to children’s social, emotional, and behavioral needs in a timely manner. Responsibilities for this are country-specific and shared across service sectors, but they generally rest with schools, at least to a certain extent [9]. Schools are considered suitable places for SEBD identification because of their direct and daily access to all children, monitoring options, and ability to involve professionals with specific expertise [10,11,12]. Indeed, children who experience SEBDs and their families frequently access support services through school [13,14,15,16]. Schools play an important role in recognizing and providing support for children at risk for, or experiencing, SEBDs.

Various methods for early SEBD identification are used in school settings, and each contains strengths and weaknesses [11,17]. *Teacher referral* has traditionally been the most common pathway for SEBD identification in schools, as teachers are well-positioned to recognize early signs of SEBDs because of their close contact with pupils and knowledge of developmental trajectories. Although teacher referral offers opportunities to identify SEBDs at an early stage, this ad hoc identification is also prone to teacher bias [18,19,20]. In interpreting pupils’ behaviors and needs, teachers are inclined to over-identify pupils from lower socio-economic backgrounds and those expressing externalizing behaviors (e.g., aggressive or disruptive behavior), while under-identifying those who are at-risk but do not express obvious signs of SEBDs (see also [17,21,22]). Teachers often feel ill-equipped for this identification responsibility [23,24,25,26,27,28], and studies suggest that they, together with other frontline gatekeepers, identify fewer than 20% of children with SEBDs [16,29,30]. Alternatively, several systematic school-based identification methods have received attention in the literature, such as *universal screening* and *selective screening*, in which the emotional and behavioral health and well-being of all or at-risk pupils are assessed by means of questionnaires [31]. These identification methods have potential and may counter bias, but they are also known to yield a significant number of false positive results and appear to have limitations in practice in terms of acceptability, feasibility,(cost-)effectiveness, and sustainability [11,17,32,33]. Many schools do not (fully) implement systematic screening methods, and the assessment and identification of children at risk for, or experiencing, SEBDs frequently rest with school staff.

As there is much to be gained in terms of school-based identification, Fazel et al. [9] emphasized the need for more research on approaches aimed at identifying these children. Following national education and youth care system reforms in 2014 and 2015, respectively, Dutch municipalities and schools were encouraged to increase their focus on prevention and early identification [34], but they may interpret and delineate their tasks and responsibilities in different ways to accommodate contextual differences (e.g., school size, location, school staff education and expertise, parental levels of education, and support options in the region). Because school staff play a central role in recognizing and responding to children’s social, emotional, and behavioral needs, differences in their identification practices may impact whether and when (signs of) SEBDs are identified. In other words, Dutch primary schools are searching for effective yet feasible ways to identify SEBDs at an early stage, and their approaches are likely to differ, which may have implications for the timeliness and quality of SEBD identification.

In most countries in the Global North, there is a reasonably established practice for monitoring children’s health and well-being in (primary) school settings [35]. When it comes to identifying SEBDs, teachers generally play an important role [19,36]. This is also the case in Dutch primary education [37]. Dutch primary school classes of approximately 25 pupils are commonly taught by one teacher for an entire school year. Because of their daily contact, these teachers are expected to be able to monitor child development and well-being and to recognize worrisome behaviors in their pupils in a timely manner [36,37]. Dutch primary school teachers are trained at higher education institutions and obtain a bachelor’s degree. In general, they have completed some courses on teaching pupils with special (educational) needs, and they may have attended additional workshops/courses as part of continuing professional development.

In many countries, designated staff members are responsible for special needs care and can be involved in the case management of pupils with (possible) SEBDs. These staff members can be school nurses, school psychologists, counselors, or care workers. In Dutch primary schools, as in several other European countries, a Special Educational Needs Coordinator (SENCO) is available at every school to guide teachers in recognizing and meeting their pupils’ needs and to coordinate special needs care [38]. SENCOs are generally (or have been) experienced teachers with additional training on special (educational) needs and the implementation and coordination of care policies. Due to their position and background, SENCOs can make an important contribution to the implementation and effectuation of a school’s approach to identifying and supporting pupils with SEBDs [39,40]. As such, SENCOs may play a pivotal role in improving the early identification of SEBDs by school staff.

Additionally, in line with common practice in the Global North, schools can call on outside expertise for pupils with suspected or identified special needs. In Dutch primary schools, SENCOs can request additional expertise or support from external organizations (e.g., educational counseling or youth care) to engage professionals with specialist knowledge for supporting individual pupils or teachers in addressing specific pupil needs, including support for SEBDs.

Besides the teacher referral method and opportunities to call on additional expertise, all Dutch schools monitor children’s health and well-being by means of a two-time screening by a school physician or school nurse during their primary school period. Prior to these appointments, parents are asked to complete a questionnaire regarding their child’s emotions and behavior (i.e., universal screening). This routine screening, stipulated by the Public Health Act, is intended to provide a general impression of children’s physical and psychosocial development and to identify pupils who may be in need of support. As such, it contributes to SEBD recognition and referral [16]. This screening only takes place twice during the primary school period, however, and the screenings are approximately four years apart. Difficulties that become apparent in the time between screenings may therefore not always be noticed in a timely manner on the basis of this approach. Research has shown that an approach that combines identification methods, such as screening along with teacher referral, is most promising and allows for both individual and school-wide monitoring of children’s social, emotional, and behavioral needs [9,11,17,41].

The present study aims to support schools and their staff in making decisions about SEBD assessments, promoting more adequate identification practices and therefore more effective SEBD identification. To build on and improve current identification practices, it is essential to explore the similarities and differences in the approaches used by different schools and school staff and to determine what implications these differences may have. Our research question is: *In which consequential ways do SEBD identification practices differ between Dutch primary schools?* On the basis of our analysis of 34 in-depth semi-structured interviews with professionals, who reflected on their experiences in approximately 80 schools, we showed that identification practices differed between schools in terms of the frequency of observations, the maintenance of the four-eyes principle, and the use of specialist knowledge, which we argue has consequences for the timeliness and quality of SEBD identification. As Dutch practices aimed at identifying SEBDs are similar to those in many other countries in the Global North, we expect the outcomes of this study on the (implications of the) differences in identification practices to also be of interest to many other countries.

## 2. Materials and Methods

### 2.1. Study Design and Sampling

This study was part of a research project examining the characteristics and effects of primary school approaches for the identification of, and support for, children with SEBDs in the Netherlands. To gain insight into the ways in which SEBD identification practices by school staff in Dutch primary schools differ, we opted for a qualitative examination. *In-depth* interviews allowed us to determine which identification procedures and practices were more or less common in schools, to explore the extent to which ‘official’ procedures and actual practices corresponded, and to identify which elements could be important for increasing our understanding of SEBD identification effectiveness. Interviews were chosen because observations would not have been feasible; professionals generally do not know in advance whether they will be dealing with a (possible) case of SEBD on a given day and whether or which identification practices will be carried out. More specifically, we opted for *semi-structured* interviews, as these allowed us to include important topics/questions in all interviews (e.g., common practices and potential differences) but would also leave room for elements that we had not considered beforehand. The research was granted approval by the Ethics Committee of the Faculty of Social Sciences (Radboud University Nijmegen; reference number: ECSW-2019-102).

We purposefully selected schools and professionals to compose a sample in which a variety of practices in schools, as carried out and experienced by different professionals, could be highlighted [42]. We selected a number of schools throughout the Netherlands that varied in terms of location and size. For each partaking school, we included professionals in different roles to allow for a triangulation of sources and to ensure that we could capture multiple perspectives on the process of SEBD identification within the same school [43]. All professionals (had) worked at several schools and were therefore able to draw on a broad range of experiences. In addition to their experiences with the nine schools we had selected, where they worked at the time of the study, each of the participating professionals shared experiences based on memories about (an average of two) other schools. We therefore obtained information on identification practices in approximately 80 primary schools in total. We continued data collection until we reached data saturation, at which point an additional interview did not yield relevant new information regarding school identification practices.

Our final sample consisted of 34 educational and clinical professionals. Among them were eleven SENCOs, eight teachers, four principals, and one school unit leader, as well as ten support providers. We refer to this group of care professionals as ‘support providers’ because they all provide additional support for children with SEBDs and school staff, despite their diverse backgrounds and functions/roles. These professionals were from different organizations (including school social workers, educational counselors, youth care workers, and a youth nurse). This sample of professionals was involved in SEBD identification in the Dutch primary school context to varying degrees and was representative in terms of age, educational background, levels of experience, and sex (94% identified as women). We obtained informed written consent from all professionals prior to their participation.

### 2.2. Data Collection

From April 2020 to November 2021, in-depth semi-structured one-on-one interviews, approximately 1–1.5 h each, were conducted by two PhD candidates, including the first author, and three other graduate students in the final stages of their education in the field of pedagogics. All interviewers received interview training prior to conducting the interviews and participated in peer meetings for exchange and alignment purposes during the data collection process. The interviews took place digitally via an online platform of the interviewee’s choice (e.g., Zoom or Microsoft Teams).

All interviews started with the interviewer describing the present study and requesting information about the interviewee’s current position, educational background, and work experience (e.g., years, positions, and employers). The informants were then asked to describe a typical working day that involved the identification of, or support for, SEBDs as well as an ideal working day (i.e., what they would prefer to do differently). In the remainder of the interview, the interviewers invited the interviewees to describe the identification practices they had applied, using a few representative cases of pupils with (suspected) SEBDs, as well as cases where they felt that identification had or had not gone well. Through their questions, the interviewers attempted to get an idea of the identification practices at the schools in which the informants worked, the informant’s specific methods, and their preferences and opinions on the effectiveness of different practices. The case descriptions made it possible to compare early identification practices and differences between schools, stakeholders, and SEBD cases. Interviewers asked for further details or clarification (e.g., ‘What was that decision based on?’ or ‘Can you give an example?’) or posed questions to determine whether certain practices could be characterized as standard or unusual (e.g., ‘Is that how it usually goes at this school?’ or ‘In which ways are your experiences at this school different from experiences at other schools in which you have worked?’). After a few cases had been discussed or the available time had passed, the interviews were concluded by asking if the informant wanted to share anything else on early identification that had not yet been mentioned.

### 2.3. Analysis

Interviews were audio-recorded, transcribed verbatim, then edited to remove personal information and replace names with pseudonyms. The first and second authors, MJ and ER, then thematically coded the interview data using a phase-by-phase recursive process [44,45] in MAXQDA V20.4.0. First, the full transcripts were read to become familiar with the data. Next, MJ initially used descriptive codes to summarize the content of specific quotes (e.g., ‘support provider involved’, ‘different stakeholders and interests’, or ‘course of action’). MJ and ER then discussed and reflected on the data in several rounds to identify patterns and distinguish themes, guided by literature on which practices could be relevant for early SEBD identification outcomes. Together, MJ and ER assigned interpretative codes (e.g., ‘regular observation’ or ‘complementary view’) and MJ attached memos to the codes that captured lines of reasoning regarding the codes’ interrelationships. We identified three major themes in which specific practices in schools differed, each potentially relevant for SEBD identification outcomes. Finally, all three authors discussed, reviewed, and refined themes until they reached a consensus.

## 3. Results

Our analyses illustrated that differences in early SEBD identification practices in Dutch primary schools largely derive from the ways in which school staff members observe and assess possible SEBD cases. We found that observation and assessment practices differ in three main elements: the frequency of observations, ensuring a four-eyes principle, and utilization of specialist knowledge. In the following, we first describe commonalities in the implementation of early SEBD identification approaches and subsequently illustrate how practices in schools differed in the three identified elements and the implications this had for SEBD identification.

### 3.1. Common Ground in SEBD Identification Practices

Above all, the number and diversity of cases cited by our informants showed that the school setting is indeed suitable for identifying SEBDs. During the interviews, one specific theme emerged repeatedly and was acknowledged by all informants as being essential to early SEBD identification: *Who is held responsible for observing and assessing potentially worrisome child behavior?* All informants indicated that the observation and assessment of possible behavioral deviations or distress are necessary to identify SEBDs. Moreover, they all agreed that both teachers and SENCOs bear a great responsibility in these ‘observational practices’; however, the point at which SENCOs or other professionals got involved differed between cases and schools. We therefore took a more in-depth look at how SEBD identification was generally implemented and the specific elements in which practices varied.

We found two early SEBD identification approaches to be common in primary school practice, with most schools adopting one or both: a stepwise approach of observation and assessment and regular observations by the SENCO. The stepwise approach involved a gradual increase in expertise utilized for observation and assessment. In this approach, teachers were expected to be the first to notice possible SEBDs. Depending on their initial assessment, others with more specific knowledge could become involved to assess the needs in a particular case. Usually, teachers approached the SENCO to request their involvement, and they then decided jointly whether the involvement of another professional was required. In almost all interviews, the informants mentioned various support providers as subsequent observers, including team members such as teaching assistants or school-employed care workers as well as professionals from external organizations. The added value of their observations was seen in their complementary nature and in their expertise. As for the second approach, the majority of our informants mentioned observations by SENCOs without initial teacher concerns as another common procedure. These classroom observations by SENCOs took place at predetermined times during the school year according to policy agreements.

In both approaches, the SENCO plays an important role. Most informants indicated that the responsibility for early SEBD identification should also rest largely with SENCOs, as their observations and assessments put teachers’ perceptions into perspective. This was expressed as follows by SENCO and teacher Dina:
“*Some teachers get on very well with a child and other teachers, who cannot keep order, may have more trouble with that child. (…) the SENCO has the ‘helicopter view’; that’s what makes [the role of] the SENCO so important.*”(Dina—experience: 18 years teaching, 3 years SENCO)

Dina’s quote reflects a notion commonly shared among our informants that SENCOs are more ‘detached’ from the classroom situation. Because of this, SENCOs can take specific classroom or teacher characteristics into account in their assessments. Moreover, whereas teachers in the Netherlands usually get a new class every year, SENCOs are responsible for pupils over the years. This ‘helicopter’ view allows for assessments of specific child behavior in comparison with other cases (e.g., concerning a different child, teacher, or approach) as well as comparisons of the same child over consecutive years. In addition, the majority of our informants believed that SENCOs can generally assess child behavior more proficiently than teachers. SENCOs tend to have more specific training and knowledge and typically do not teach; their responsibility is to consider the needs of pupils and teachers. Teachers also described that the SENCO “*gives advice*” and “*knows how to proceed*” in cases with special needs. Our informants thus assumed that SENCOs are able to observe, provide independent assessments of child behavior, and arrange the input of professionals with more extensive SEBD knowledge or experience as needed.

While the above might indicate uniformity in observational practices, our informants’ experiences with individual schools and SENCOs pointed to differences between settings. We found that different ‘modes of observation’ were used, ranging from minimal to thorough in terms of when and how frequently observations took place, whether and how independent observers were included, and the extent to which specialist knowledge was used. Below, we illustrate how practices differed among schools and SENCOs in terms of the three identified elements.

### 3.2. Frequency of Observations

Our informants had varying experiences of whether and how frequently certain professionals observed child behavior. Their experiences mostly related to the practical limitations of teacher observations and the potential of scheduled classroom observations by SENCOs.

Although Dutch teachers typically see children in their classroom almost daily, our informants, including teachers, emphasized that teachers are restricted in their ability to perform focused observations. As SENCO and teacher Vera told us:
“*They [teachers] have so much to do nowadays, that sometimes they almost forget to observe the children.*”(Vera—experience: 37 years teaching, 6 years SENCO)

Vera questioned whether teachers have the opportunity to observe individual pupils for potential SEBDs because they have other priorities, and their main focus is on teaching. Educational counselor and special education teacher Brenda asserted that teachers are therefore unable to notice every SEBD signal early on: “*as a teacher you just can’t see everything; that’s impossible. (…) You’re just missing signals*”. In addition, our informants indicated that teacher observations are restricted, as most Dutch teachers get a new class every year, which prevents them from monitoring pupils over consecutive years. Leaving this responsibility for recognizing possible SEBDs primarily with teachers could thus perpetuate inadequate identification.

To overcome these restrictions, several informants underscored the relevance of regular, scheduled observations as well as observations by SENCOs for recognition and monitoring purposes. Most schools have drawn up policy agreements (i.e., the so-called ‘care structure’) in which these options are combined that specify regular moments during the school year when the SENCO visits classes to ensure pupils and possible difficulties are observed on a regular basis over time. Brenda said about this:
“*And if nothing seems wrong, then you will be done with that child within half a minute, but the fact that it [classroom observation] is scheduled forces you to at least keep an eye on it [child’s condition].*”(Brenda—experience: one year educational counselor and special education teacher, academic level)

In her opinion, working according to a schedule should guarantee the frequent observation and careful monitoring of each pupil’s condition over time. In the interviews, subsequent teacher–SENCO discussions of individual pupils were also mentioned. Depending on a minimum frequency—Brenda referred to twice a year—this combined system of planned observations and pupil discussions should counteract identification inadequacies, such as teachers missing signs of SEBDs or unduly assessing behavior as problematic.

Multiple informants labeled this way of working as important, but in practice, the adherence to scheduled observations appeared to differ. Whereas some teachers indicated that the SENCO regularly visited their classroom to observe, SENCO Jantine admitted that “*I actually never get around to that*”, and Brenda noted that some SENCOs prioritize other matters. As a consequence, signals could remain unnoticed or unmonitored, with the risk of persisting “*year after year*”, as Brenda said. SENCO observations are thus seen as having the potential to improve early identification, provided they are conducted regularly and frequently.

When it comes to observational monitoring, SENCOs engaged in regular observations to varying degrees. Whether observations were scheduled and conducted frequently seemed to depend both on school policies/regulations and on choices and priorities of SENCOs. Consequently, different observational practices affected the frequency with which children were monitored and therefore the possibilities to identify SEBDs in a timely manner.

### 3.3. Ensuring a Four-Eyes Principle

Besides differences in the frequency of observations, our informants had different experiences with complementary observations conducted by SENCOs or support providers in addition to observations made by teachers in daily practice. As described above, additional observations by different professionals could help to ensure that signals of SEBDs are picked up even if missed by teachers. More importantly, however, our informants indicated that complementary observations *by someone outside the classroom dynamics* could put teachers’ perceptions into perspective and counteract possible bias. In other words, independent complementary observations, to which we refer as the ‘four-eyes principle’, could promote a more thorough assessment in addition to contributing to the observation frequency.

We found schools and SENCOs apply this four-eyes principle differently. The above-cited scheduled observations were mentioned as one way to put this into practice, but according to our informants, independent complementary observations more commonly followed a request generally made by teachers (i.e., the stepwise approach). Most informants stated that when teachers express concerns, SENCOs should get involved to conduct such additional observations or arrange for another professional to provide an independent opinion. In this regard, SENCO Jantine explained:
“*When they [teachers] see trends, when they really start seeing signals more often, then I get involved.*”(Jantine—experience: 12 years teaching and providing special needs care, 2 years SENCO)

Many agreed on this, including SENCO and teacher Vera, who explained that teacher concerns or troubles are typically a reason for SENCO involvement because “*a SENCO usually only comes into the picture when something does not run smoothly*”.

Although it may seem obvious that SENCO involvement occurred following the expression of teacher concerns, our informants indicated that the moment when either a teacher or SENCO believes that things no longer “*run smoothly*” can vary greatly, and views on this may conflict. That is precisely why multiple informants pointed out the importance of early SENCO involvement as well as the four-eyes principle. They implied that observers are always biased; the behaviors considered worrisome can be affected by personal characteristics, such as limited or extensive knowledge, experience, or skills. Furthermore, assessments can be affected by situational aspects, such as the level and relative presence of SEBD cases in a particular class. Timely complementary observations may then contribute to a more valid assessment and identification of SEBDs.

Some SENCOs were naturally involved with teachers and pupils and were quick to address teachers’ concerns by conducting or arranging additional observations:
“*Then I thought, ‘Something needs to be done about this’. There always is a SENCO in school, who often comes to observe the classroom for a while, after which any issues are discussed: What could we arrange?*”(Lonneke—experience: 10 years teaching)

Teacher Lonneke reported that this SENCO relied on her initial assessment and quickly provided complementary views. SENCO Marit shared her similar experiences of being responsive to teachers and getting involved early:
“*That [my early involvement] was because the teacher actually gave a signal already after the first day, like: ‘I have someone in my classroom now, Marit, and they do not show an average development (…). I have seen things that immediately raised questions.’ So, that teacher got me involved.*”(Marit—experience: 15 years SENCO, including 10 years in special education)

Marit indicated that teachers sometimes perceive signals of worrisome behavior early on, to which SENCOs should quickly respond. The above case involved exceptional behavior, suggesting that the type of behavior and visibility play a role in recognizing possible SEBDs. Multiple informants mentioned that externalizing behaviors (e.g., aggression or defiant behavior) are experienced as much more disruptive and therefore reported earlier by teachers, while internalizing behaviors (e.g., withdrawn behavior or depression) are more easily overlooked. This highlights the importance of a four-eyes principle to prevent certain child behaviors from being unnecessarily problematized and others from being disregarded.

Teacher Lonneke, who had worked at as many as nine schools in both regular and special education, indicated that she experienced “*enormous differences*” about when a four-eyes principle was met by the SENCO. Indeed, the interviews demonstrated that high degrees of responsiveness were not common for all SENCOs. Despite an established ‘care structure’ and SENCOs’ intentions to get involved when teachers “*start seeing signals*”, teachers in particular reported that their concerns were not always (directly) addressed. Some of them indicated that low responsiveness resulted in them having to approach their SENCO repeatedly, which delayed the complementary observations and allowed worrisome behavior to persist or worsen in the meantime. This occasionally made teachers reluctant to share concerns with their SENCO at all, preventing a timely independent assessment.

With a view to these differences in responsiveness, Brenda explained how schools’ care policies could have an impact on the course of the identification process:
“*I also think that the care structure within the schools should simply be better organized. Not in all schools, but there are schools where this is not yet sufficient. As a result, some signals are known to teachers year after year, but only when things really get stuck (…) is a signal of concern put in place [by the SENCO].*”(Brenda)

Brenda experienced that the care structure that should delineate processes and responsibilities on monitoring, identification, and support was poorly organized in some schools. Policies were not available, not fully known, or not adhered to. Consequently, school staff members acted inconsistently, which hindered the timely involvement of other professionals. For example, Brenda described situations in which the SENCO only became involved or requested another professional to assess the situation when a pupil’s behavior got out of hand or when it was realized that a pupil would be unable to move on to regular secondary education.

Besides the implementation of and adherence to schools’ care policies, several informants implied that school management decisions or local regulations could affect SENCOs’ abilities to ensure a timely external complementary view. They described how, at some schools, professionals with specialist knowledge from a mental health context became part of the school team, which benefited their accessibility. For example, some spoke of municipality initiatives where youth care workers were present at the school for early SEBD identification purposes. While in the Netherlands an extensive formal application is normally required to arrange funding for their involvement, this was not needed within these initiatives. Consequently, teachers and SENCOs said they appealed to them as expert complementary observers for SEBD identification purposes more frequently and quickly.

SENCOs’ abilities to ensure a timely complementary view were thus affected by care policies and the ease of engaging support providers, but our informants indicated these also depended on SENCOs’ individual involvement and competences, particularly their approachability and proactivity. Several teachers pointed to a lack of these traits, which they viewed as missed opportunities for SENCOs to become involved early on. Teacher Iris compared the state of affairs at her previous and current school:
“*Well, there [at the previous school] the SENCO was actually always there and very visible. (…) I don’t experience that at all with ours at the moment; they are somewhere in the back of a damn corner and you really have to go there yourself. You don’t see them walking in the corridor, so to speak.*”(Iris—experience: four years teaching, two different schools)

Iris implied that a complementary view is easier to arrange when teachers and SENCOs see each other frequently and do not have to make formal requests. In addition, visibility and proactivity from the side of the SENCO, such as walking in the corridors and asking teachers about any particular concerns, may contribute to the exchange and verification of possible SEBD signals. These proactivity and approachability traits were also reported to be important for school-employed support providers.

Apart from complementary observations conducted or arranged by SENCOs, our informants mentioned additional ways to promote a four-eyes principle and counterbalance teachers’ restrictions in their ability to perform focused observations. For example, teacher Iris also reported that at her previous school “*all kindergarten groups had a teaching assistant in the classroom every day*”*,* which in her eyes helped because ‘two people can perceive more than one’. She further described being given opportunities for observation outside the classroom context, such as “*home visits for lunch on a Wednesday afternoon*”. Furthermore, principal Jonas talked about a new approach at his school, in which the composition of classes and teachers was regularly changed. As a result, “*the teachers are so close to each other, they see each other teaching and also see the other’s children, and therefore also discuss the children together*”. Since teachers and teaching assistants are subject to the aforementioned restrictions and generally lack the training and monitoring possibilities that SENCOs have, these alternatives cannot replace regular complementary observations by SENCOs or external professionals. Still, both Iris and Jonas indicated that additional observations in different settings give more opportunities for multiple observers and provide supplementary information that may help recognize potential SEBDs at an earlier stage.

In other words, teachers experienced that the more difficult it was to get someone outside the classroom dynamics involved for complementary observations, the longer it took for child behavior to be properly assessed. By being accessible and responsive in various ways, SENCOs and support providers could facilitate the application of the four-eyes principle. In line with this, policies requiring regular class visits were found to increase SENCO approachability, thereby simultaneously contributing to more frequent observations and the four-eyes principle.

### 3.4. Utilization of Specialist Knowledge

In addition to differences in the frequency of observations and in meeting the four-eyes principle, our informants had different experiences with when and to what extent schools and SENCOs appealed to professionals with specialist knowledge for observation and assessment, either available within the school team or from external organizations.

As for specialist knowledge that could be utilized for early SEBD identification, the professionals working in the context of school-provided support had different levels and types of relevant knowledge. As a consequence, school teams differed in the level of expertise and experience available among team members (e.g., worked in few/many schools, worked in regular/special education, or any additional training). The informants clarified, however, that limited expertise within the team did not have to be an obstacle because specialist knowledge could be called upon from external organizations and could be exchanged and transferred among team members. Schools, and more specifically SENCOs, appeared to make use of these options for early identification purposes to varying degrees.

First, besides teachers and SENCOs sharing their knowledge one on one, our informants described ways in which some schools facilitated the exchange of available expertise among team members more broadly to promote observation and assessment. They described innovative forms of team collaboration, in which regular observations of each other’s pupils and ways of working were central. For example, the new group-changing approach described by principal Jonas naturally allowed teachers to see and discuss each other’s courses of action and to observe each other’s pupils from their own areas of expertise. Moreover, this facilitated knowledge transfers by team members with additional training outside (regular) education. This approach was well received by team members and proved to be particularly helpful in recognizing and discussing pupils who could need extra support. Likewise, SENCO Jantine mentioned the ‘learning square’ at her school, where at times, a number of teachers and their pupils were working in the same place at the same time. According to our informants, such approaches did have a precondition that staff wanted to contribute to the utilization of available knowledge; team members had to be open to exchange and feel supported in doing so rather than ‘judged’.

Second, in addition to drawing on team members’ available expertise, the informants indicated that almost all schools made use of specialist SEBD knowledge from professionals working or being trained outside the (regular) school context, albeit to a greater or lesser extent. They stated that, apart from an independent view, these professionals could bring a different perspective along with their specific expertise. Youth care worker Frida, who worked at several schools concurrently, said:
“*Here [in education], people look at it [child behavior and development] in such a different way. Sometimes there are just blind spots in education. (…) So [the teacher] does not receive the [implicit] signals of the request for help. And in that way, problems can simmer for a very long time without actually being noticed.*”(Frida—extensive experience in youth care, including crisis care, and in providing parenting support)

Frida felt that the school context differed greatly from the (mental health) care context in terms of what professionals mainly focus on, how development is viewed, and how child behavior is interpreted. Similarly, other informants reported that school staff are usually much more involved in the curriculum, learning performance, and work attitudes, whereas support providers are more concerned with mental health, personal development, or family situations. Accordingly, SENCOs and teachers were said to have a stronger tendency to assess whether students need support from a learning progress perspective, while support providers would be more adept at interpreting behavior. Although the interviews showed that teachers and SENCOs sometimes also had specialist knowledge, Frida indicated that support providers could additionally bring “*a different intuition*” because of their clinical or pedagogical background, which could help to identify SEBDs at an earlier stage. On the other hand, care workers repeatedly stated that their specialist knowledge and perspective contributed to recognizing apparently deviant behavior as being part of normal development. This so-called way of “*normalizing certain behaviors*” could reverse misjudgments. The informants therefore attributed added value to these outsider perspectives in terms of early identification.

We found differences in schools’ utilization of such specialist SEBD knowledge from professionals outside the school context. More specifically, we found that practices differed for SENCOs’ *timing* of appealing to these professionals. Although teachers and SENCOs said they were usually willing to jointly decide on ‘outsider’ involvement, the moment at which they deemed such involvement necessary could differ. Some informants reported that their SENCO rather quickly requested (external) professionals with more specialist knowledge to shed light on suspected SEBDs. Such timely expert views generally made teachers feel supported because they knew what was going on and how to respond appropriately. Other teachers and support providers felt that SENCOs sometimes waited too long before requesting an expert view.

Informants mentioned possible reasons for these differences in the timing of calling on additional knowledge, which we will touch upon briefly. To begin, the moment of requesting expertise appeared to be partly dependent on the type of SEBD and accompanying behavioral expression. Informants reported externalizing behavior (e.g., aggressive or disruptive behavior) to be more likely than internalizing behavior (i.e., withdrawn behavior or depression) to cause tension and distress not only for the pupil who expressed the behavior but also for the teacher and the class concerned. SENCOs were then inclined to call on specialist knowledge more quickly. Still, teacher Lonneke described a situation where her SENCO refused to involve someone with more specialist knowledge, while a pupil frequently displayed worrisome aggression. This made Lonneke feel discouraged, strained, and not taken seriously in her assessment. In this regard, yet not in this example, an educational counselor noted that teachers occasionally bypassed their SENCO and tried to arrange expertise on their own if they felt the SENCO fell short in meeting their or their pupils’ needs.

Furthermore, the interviews showed that the moment of appealing to external professionals depended on the level of knowledge and persistence of the SENCO. While SENCOs typically applied their own expertise before calling on other professionals, a few informants indicated that this could also hinder the utilization of specialist knowledge. For example, educational counselor Mirthe clarified that a SENCO’s intermediate level of knowledge could complicate or delay the decision to engage expertise:
“*The SENCOs did so much themselves in terms of advising and the teachers went on for so long, that by the time we arrived, they were actually already here [gestures that limits were reached] and were no longer open to advice. (…) It is more feasible to ask for help earlier so that people feel space, and then you can use the available expertise.*”(Mirthe—extensive experience in education as teacher, SENCO, and in management and educational development, five years educational counselor)

At this school, educational counselors recurrently became involved at a late stage. As a result, school staff lost confidence in using their expertise and trying different approaches. Mirthe indicated that the SENCO’s persistence, although commendable, could undermine their own commitment.

Lastly, SENCOs’ timing of calling on additional expertise appeared to depend on the accessibility of support providers. As pointed out above, informants explained that local regulations and initiatives, as well as the composition of an individual school’s team, could result in support providers being part of the school team or being present at school on a regular basis. This greatly benefited the utilization of specialist knowledge for SEBD identification because formal administrative hassles were eliminated.

Taken together, schools appealed to specialist knowledge for observation and assessment to varying degrees and in different ways. Opportunities for knowledge exchange among team members and the timely involvement of expert professionals were found to have several potential advantages. These approaches made most teachers and SENCOs feel that they were taken seriously by substantiating their initial assessments, and they provided the necessary expertise for SEBD identification at an earlier stage. Moreover, observations by professionals from outside the school context contributed to the utilization of specialist knowledge while also meeting the four-eyes principle. In addition to facilitating a better and earlier understanding of possible SEBDs, the timely involvement of expert professionals could also benefit objectivity.

## 4. Discussion

This study was conducted to explore current early SEBD identification practices by primary school staff and to understand how differences in these practices may affect the quality and timeliness of SEBD identification in pupils. Using semi-structured in-depth interviews with educational and clinical professionals, we found that identification practices in regular primary schools differed in three main elements: the frequency of observations, the extent to which a four-eyes principle was ensured in assessments, and the utilization of specialist knowledge. At some schools, for example, children were systematically and frequently observed, including by professionals outside the classroom dynamics with high levels of specialist knowledge. At other schools, few or irregular complementary observations were carried out. Differences in the three main elements of identification practices likely impact the likelihood of SEBDs being recognized in a timely and ‘unbiased’ manner. Observation and assessment by SENCOs or support providers in particular have added value, as they provide a complementary view of teachers’ perceptions and utilize specific expertise. This is relevant because research shows that children with difficulties are more often overlooked in teacher nominations than in screening methods, especially those without apparent signs of SEBDs [17,21,22,46,47,48,49]. Ensuring frequent observations, not only by teachers but particularly by observers with specialist knowledge who provide an independent complementary view, may therefore promote earlier recognition and a more expert assessment.

Consistent with research in other countries, our informants described SEBD identification as a shared responsibility between teachers, other school staff, and (external) support providers [24,50,51]. Teacher referral is the norm in the Dutch education system, and the SENCO was consistently described as a key figure in the identification process. Indeed, we found that the different identification practices in schools largely resulted from differences in SENCOs’ practices. These ranged from SENCOs who were barely seen in the classrooms or corridors and not very responsive to teachers’ concerns about specific pupils to SENCOs who were proactive, frequently present in and around classrooms, sensitive to perceived concerns, and quick to call on expertise. Our study showed that these differences in SENCOs’ identification practices can either ensure or compromise frequent, complementary observations and the utilization of specialist knowledge. The SENCO role—or a similar role in which someone with specific expertise (in the school but outside the class dynamics) is involved in the early identification of possible SEBDs in pupils—is a crucial prerequisite to ensuring all three elements of SEBD identification. Studies have shown that SENCOs interpret and enact their role and tasks differently [52,53], and it would therefore be important to study the reasons underlying the different identification practices used by SENCOs. Our findings point to two recurring themes that may be worth examining in this regard. First, professionals indicated that policies on procedures and services could guide common practice in schools to some extent (e.g., agreements on regular classroom observations by SENCOs or arrangements about the availability of support providers). Second, practices appeared to depend on personal factors, such as the personal traits and views of SENCOs. More research is needed.

### 4.1. Practice Implications

Our findings suggest that when the observational practices of school staff meet the three identified elements (i.e., frequent observations, complementary views, and use of specialist knowledge), they may contribute to early SEBD identification. We also found that the ultimate effectiveness of an approach depends on how responsibilities are allocated and whether appropriate preconditions are provided. In line with this, the insights from our study indicate that school teams can undertake two key actions to improve early SEBD identification efforts and outcomes in Dutch primary schools.

First, schools should facilitate consultations for teams to discuss how their identification practices can accommodate the three elements. In order to appropriately address SEBDs, clarity on professional roles and tasks, as well as the coordination of practices, are needed [50]. As was reported in the literature, our analysis showed that school staff generally agree that identifying and responding to child SEBDs is part of their professional role(s), but their views diverged as to exactly what their own involvement should entail [24,50,54]. School teams should therefore deliberate about their current and preferred identification procedures and practices (e.g., stepwise approach and scheduled observations), and discuss how these can be better aligned and supported. In this way, schools may promote more frequent observations and the timely involvement of SENCOs and other professionals. As an example, a flow chart indicating when additional expertise should be requested or called upon can clarify both who is to arrange accessing a complementary view or specific specialist knowledge and at what stage. Furthermore, when team members agree on identification method(s) and SENCO involvement, SENCOs will be better able to coordinate practices.

Second, school management and SENCOs should look for ways to optimally utilize SEBD expertise. Our findings imply that specialist knowledge utilization may contribute to more accurate child behavior assessments, and schools have several options to promote this; for example, schools could involve external experts more often or more intensively. Additionally, schools could invest in interprofessional collaborations to promote not only the utilization of but also the exchange of relevant knowledge and skills for SEBD identification [55,56,57]. Furthermore, schools may consider (whole-school) staff training to build team competence and empower staff members to better recognize and respond to possible cases of SEBDs [11,19,24,25,36]. In other words, schools are encouraged to invest in ways to enhance the utilization of expertise to help school staff provide timely and appropriate support for pupils with SEBDs [36,58].

### 4.2. Limitations, Strengths, and Future Directions

In conducting the present study, we met the standards for this type of qualitative research, for which some limitations are inherent. The professionals who took part in this study were self-selected, and therefore constituted a rather small sample of professionals willing to speak about their personal experiences with school-provided support for children with SEBDs in the Netherlands. Despite this, a significant proportion of our informants had experience not only in multiple schools but also in different positions/roles (e.g., SENCOs who used to be teachers), which facilitated comparisons of experiences by the same person and also contributed to the comprehensiveness of the data. Furthermore, as educational and mental healthcare systems and endeavors vary by country, the differences in identification practices found in this study do not automatically apply to every school-provided support setting. Nonetheless, our reflections on different practices in Dutch primary schools indicate that certain aspects are important for (developing) early SEBD identification approaches that (partly) rely on teacher referral. The three elements we distinguished (i.e., frequency, complementary views, and specialist knowledge) are relevant to school and healthcare systems internationally because the Dutch situation shows similarities with SEBD identification in other countries. Moreover, our findings point to the contributions that various professionals can make to early SEBD identification; for example, SENCOs play a major role in coordinating and carrying out identification practices, which more generally indicates the added value of having someone on the school team with specific expertise, dedicated responsibility, independence, and the ability to call for additional support. Together, this suggests that early SEBD identification would benefit from an examination of the structure of education and care systems to determine which types of professionals with which competencies should be in which positions.

To promote the implementation of (more) effective early SEBD identification approaches, a better understanding of where different practices come from would be beneficial. In Dutch education, the three elements we revealed are commonly part of the identification approaches in schools to some extent; however, teacher referral is interpretated and implemented differently. As SENCOs were found to be key figures, additional research could examine the reasons underlying the differences in their role fulfillment in this context. The present study suggests that various factors are likely to play a role (e.g., school policies, personal characteristics, and type of difficulties) (see also [59]). More research on these factors is needed to allow for a more fruitful implementation of practices in line with the three elements. Ultimately, a greater understanding of the reasons underlying the differences in identification practices and the preconditions for effective practices will facilitate an improved SEBD identification process, which should lead to more children receiving timely and appropriate support.

## 5. Conclusions

This study revealed that, although the school setting is basically well suited for early SEBD identification, differences in staff practices in terms of the frequency of observations, complementary views, and specialist knowledge utilization have potential consequences for the timeliness and quality of SEBD identification. Our findings imply that early SEBD identification can be improved by tailoring schools’ approaches while investing in more frequent observations and obtaining complementary views from expert professionals. As key figures, SENCOs can play an important role in this respect.

## Data Availability

The data presented in this study are not publicly available because they cannot be completely anonymized. We have specified the protection of the interview data on the participants’ informed consent forms.

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
