# Peer review of "Early Identification of Social, Emotional, and Behavioral Difficulties by School Staff: A Qualitative Examination of Dutch Primary School Practices"

_ijerph, 2022, doi:10.3390/ijerph20010654_

Round 1

Reviewer 1 Report

It was such a pleasure to review this well-presented work qualitatively examining the current practice of early SEBD identification across Dutch primary schools. The manuscript is very well-written and the methodology is rigorous and adequate. One suggestion for the authors is to include IRB-related information in the main text under Materials and Methods. 

Additional minor comments for the authors' reference: 

  1. Line 9 - duplicated “Correspondence”
  2. Line 148 - duplicated “in”
  3. Line 403 - please confirm “nine” vs “9”

Reviewer 2 Report

GENERAL COMMENTS

The aim of this paper was to examine the differences in identification practices by school staff and their potential implications for early identification effectiveness. Although this article addresses an interesting topic, many issues should be addressed before publication.

SPECIFIC COMMENTS

INTRODUCTION
The introduction needs major revision and clarification. First, the aim of the study is not clear. Also, the way the authors build up their introduction does not lead to the research question. Although much of the necessary information regarding the background is already briefly written down, the authors should re-structure their introduction, explaining why their research is important. Why SEBD identification in Dutch primary schools is different from other countries? Why use SEBD? Why the qualitative study is preferred to the quantitative study? Thus, it is recommended that the authors expand this part: how SEBD could improve the early identification of social, emotional and behavioural difficulties by the school staff. More importantly, this should lead to a clear research question – the current research question is not specific.

METHODS
The methods section needs major revision. I am very confused. Is this a salami-slicing manuscript? Why is it part of the larger research project? Is the study published before? Which part has been published?  

2. Please provide an ethical approval code for dealing with human participants and vulnerable populations.

3. Please include sample size calculation. I think the sample size is too small. Is the data reached saturation?

4. Are the participants all males or females? Not mention. Do educational and clinical professionals have the same capacity to represent the population of the sample?

5. Who conducted the interview? Is the interviewer well-trained? How many interviewers involve?

6. Who involves in analysing the data? What is the level 1, 2 and 3 coding?

7. Please provide a reference on the analysis method and steps.

RESULTS
Sufficient.

DISCUSSION
In the discussion section, the authors should further discuss their findings and the implication of these findings. They should also discuss their findings in more depth. The authors also discuss many topics that are not related to the results. In addition, they describe many studies in great detail, which is not necessary for the discussion. The limitation of the study is missing. Please revise.

Thank you.
